# High-Flux Ultrafiltration Membranes Combining Artificial Water Channels and Covalent Organic Frameworks

**DOI:** 10.3390/membranes12090824

**Published:** 2022-08-24

**Authors:** Kai Liu, Jinwen Guo, Yingdong Li, Jinguang Chen, Pingli Li

**Affiliations:** 1School of Chemical Engineering and Technology, Tianjin University, Tianjin 300072, China; 2Tianjin Key Laboratory of Membrane Science and Desalination Technology, State Key Laboratory of Chemical Engineering, Tianjin University, Tianjin 300072, China

**Keywords:** high-flux membranes, artificial water channels, COFs, composite structure

## Abstract

Artificial water channels (AWCs) have been well investigated, and the imidazole-quartet water channel is one of the representative channels. In this work, covalent organic frameworks (COFs) composite membranes were fabricated through assembling COF layers and imidazole-quartet water channel. The membranes were synthesized by interfacial polymerization and self-assembly process, using polyacrylonitrile (PAN) ultrafiltration substrates with artificial water channels (HC6H) as modifiers. Effective combination of COF layers and imidazole-quartet water channels provide the membrane with excellent performance. The as-prepared membrane exhibits a water permeance above 271.7 L·m^−2^·h^−1^·bar^−1^, and high rejection rate (>99.5%) for CR. The results indicated that the composite structure based on AWCs and COFs may provide a new idea for the development of high-performance membranes for dye separation.

## 1. Introduction

Currently, water shortage has become a serious problem due to human activities and poor water treatment [1]. Membrane-based water purification technologies have been frequently studied due to their simple process, high efficiency, and low energy consumption [2,3]. Both high selectivity and excellent permeability are required for the ideal separation membrane.

As the most important component of biofilms, the channel aquaporin (AQP) plays a crucial role in the unique exchange of particles between cells and their environment. Owing to its high water permeability and superior selectivity [4,5], AQP with proteo-liposomes has been applied in the fabrication of separation membranes [6]. However, the stability of AQP under harsh conditions and compatibility with substrate membrane remains a challenge to be solved [7].

Researchers have tried to mimic the structure of AQP in recent years. AQP has a typical hourglass structure with high enough filtration selectivity to transport water through electrostatic repulsion in the aromatic and arginine (ar/R) constriction region [8]. Owing to the reduction in the collective hydrogen bonding, the positive charges at the entrance of the AQP channel could further improve the water transport activity [9,10]. This distinctive structure of AQP provides an idea for the synthesis of AQP alternatives. As one possible substitute, artificial water channels (AWCs) can provide an effective pathway towards membrane by mimicking AQP’s mechanism. AWCs, reported by Barboiu in 2011 [11], are synthetic water channels with self-assembled columnar structures. The central hydrophobic or hydrophilic pore guarantees the directional transfer of water, and the outer hydrophobic exterior matches the polymeric membrane environment.

The biomimetic hydrophobic AWCs have been investigated over the past 30 years. Ultrashort carbon porins (CNTPs) were inserted in a lipid bilayer by Noy et al. [12]. Gong et al. successfully synthesized a series of (m-phenylene-ethynylene) m-PE macrocycles, and their permeability could reach 4.9 × 10^7^ H_2_O·s^−1^·channel^−1^ [13]. Hydrazide-appended pillar arene was first introduced by Ogoshi et al. in 2008 [14]. Hou et al. synthesized a second generation of pillar arene PAP layer and applied it in the transmembrane transport of chiral amino-acids with a permeability of 3.5 × 10^8^ H_2_O·s^−1^·channel^−1^ [15]. Moreover, the imidazole-quartet (I-quartet) channels obtained by self-assembly of alkyl ureido-imidazole (HC6H) [11] was investigated to improve membrane permeability without sacrificing its selectivity. However, low compatibility with the substrate needs to be addressed.

Recently, covalent organic frameworks (COFs) have attracted a lot of attention in the fields of energy-storage, catalyst, and separation. As a new class of crystalline organic porous material, COFs are composed of H, B, C, O, N, and other light atoms, and possess unique properties, such as inherent porosity, good pore aperture, and abundant functional groups [16,17,18]. Compared with traditional polymer membrane materials, the separation efficiency of COF membranes has been greatly improved. However, there is still a trade-off problem between permeability and selectivity. Appropriate substrates have been proven to be a feasible way for the further enhancement of membrane performance [19]. Construction of a 2D+1D structure via inserting 1D cellulose nanofibers (CNFs) has been reported to improve both selectivity and permeability [20]. In addition to the above methods, inserting specific water channels might be an effective way to optimize the performance of COF membranes. The well-organized pores in the COFs’ active layer provide more insertion sites for HC6H. The shielding effect of embedded channels could enhance the sieving ability of the membrane, and the multiple interaction between COFs and water channels could improve the membrane stability. Meanwhile, the selective water transport capacity of water channels can also ensure the enhancement of water permeability of the membrane without sacrificing rejection.

In this work, water channels derived from the self-assembly of alkyl ureido-imidazole (HC6H) were employed for the performance enhancement of COF membranes. HC6H is a typical self-assembled channel with an artificial tubular structure that relies on strong intermolecular hydrogen bonding to achieve a stable state (Appendix A). During the self-assembly process, HC6H can spontaneously insert into the hydrophobic region of the membrane matrix to form a cylindrical structure. Therefore, HC6H could be used as a molecular scaffold for the construction of I-quartet, stabilized by internal water wires (Appendix A) [11]. These I-quartets were then inserted into COF (TpPa) membranes and connected by hydrogen bonding to form a stable composite membrane structure, as shown in Figure 1. Its separation performance was investigated and optimized in term of water permeance, dye rejection, and stability. The results showed that the composite COF membranes with high permeance and selectivity exhibited great potential in the development and industrialization of high-performance membranes.

## 2. Experimental

### 2.1. Materials

Polyacrylonitrile (PAN) ultrafiltration membranes (mean pore size: 0.134 µm) were provided by Shandong Megavision Membrane Engineering and Technology Co., Ltd. (Shandong, China). P-phenylenediamine (Pa, 99.9 wt%) was obtained from Innochem (Beijing, China). 1,3,5-triformylphloroglucinol (Tp, 95 wt%) was supplied by Bide Pharmatech Ltd. (Shanghai, China). Hexamethylene diisocyanate (HDI, 97 wt%) and histamine (96 wt%) were obtained from McLean Biochemical Technology Co., Ltd. (Shanghai, China) Tetrahydrofuran (THF, 99 wt%). N, N-dimethylacetamide (99.5 wt%), ethylacetate (99.5 wt%), and acetonitrile (99.5 wt%) were purchased from Kemio Chemical Reagent Co., Ltd. (Tianjin, China). Acetic acid (99 wt%), n-hexane (99 wt%), and sodium hydroxide (NaOH, 98 wt%) were obtained from local suppliers. Dyes including methyl blue (MB), congo red (CR), acid fuchsin (AF), chrome black-T (CB-T), and rhodamine B (RB) were provided by Aladdin (Shanghai, China).

### 2.2. Preparation of HC6H

The synthesis of HC6H is shown in Appendix A. Hexamethylene diisocyanate and histamine (molar ratio of 1:2) were poured into a mixture of THF (5 mL), N,N-dimethylacetamide (10 mL), and ethyl acetate (5 mL). After ultrasonic treatment and shaking for 10 min, the above mixture was heated at 12 °C for 15 min and acetonitrile (5 mL) was added. The mixture was then heated at 120 °C and stirred at reflux for 1 h. Finally, the product was cooled at room temperature for 3 h. It was then washed with THF and filtrated to obtain a white powder, which was vacuum dried for 10 h.

### 2.3. Synthesis of Composite Matrix Membrane

Figure 2 illustrates the fabrication process of the composite matrix membranes by interfacial polymerization (IP). Firstly, HC6H powder was dissolved in anhydrous ethanol and sonicated for 30 min to obtain a HC6H solution. The solubility of HC6H is confirmed as 0.42 ± 0.01 g per 100 g ethanol at room temperature. The PAN ultrafiltration membrane was hydrolyzed in sodium hydroxide solution (1 mol/L) at 60 °C for 1.5 h, and then immersed into HC6H solution. Pa (0.2 wt%) and Tp (0.02 wt%) was dissolved in deionized water and n-hexane, respectively. The 25 mL Pa solution (including 250 µL acetic as the catalyst) was poured onto the active surface of the PAN substrate and underwent adsorption for 30 s. After that, the excess Pa solution was removed from the substrate and then a 25 mL Tp hexane solution was added onto the Pa-saturated substrate. When the n-hexane solution came into contact with the surface of the Pa-saturated substrate, the color of the membrane surface immediately changed to yellow. After 30 s of reaction, the residual solution was drained off, and the membrane was heated at 60 °C for 5 min to form a more stable hexagonal framework. Finally, the membrane was transferred to deionized water for 3 h before usage. 

### 2.4. Membrane Characterization

The chemical structure of HC6H powder was explored using Fourier Transform Infrared Spectra (FTIR, IRAffinity-1S, Marlborough, MA, USA) in the scanning range of 500–4000 cm^−1^ and nuclear magnetic resonance spectrometry (NMR, JEOL JNM ECZ600R, Tokyo, Japan). Thermal stability of HC6H powder was determined by a thermo gravimetric analyzer (TGA, Netzsch TG 209F3, Karlsruhe, Germany).

The morphologies of the mixed-dimensional membrane were characterized by a scanning electron microscopy (SEM, Regulus 8100, Tokyo, Japan). TEM images of the composite matrix membrane were obtained using field emission transmission electron microscope (TEM, JEM-1400Flash, Tokyo, Japan). Atomic force microscopy (AFM, Dimension icon, Karlsruhe, Germany) was used to detect surface roughness.

Membrane surface hydrophilicity was analyzed through water contact angle measurer (WCA, OCA15EC, Silicon Valley, CA, USA). The water contact angle measurement of each sample was repeated for five times to obtain the average value. X-ray diffractometer (XRD, D8-Focus, Karlsruhe, Germany) was used to explore the interlayer spacing of the powder and composite matrix membranes in room temperature at the range of 5° < 2θ < 30° with a step of 0.02°/s. Elements and functional groups of the composite matrix membrane were obtained from the X-ray photoelectron spectroscopy (XPS, Kratos, Manchester, UK) and Fourier transform infrared (FTIR, IRAffinity-1S, Marlborough, MA, USA) spectra with the range of 400–4000 cm^−1^. The UV/V spectrophotometer (UV3600 Shimadzu, Tokyo Japan) was used to analyze the concentrations of dyes in feed, permeated, and retentate solutions.

### 2.5. Membrane Separation Performance

Dye rejection and water permeability of the composite matrix membranes were determined by a cross-flow filtration system with an effective membrane area of 28.26 cm^2^. Before the test, the membrane was pre-compacted under 2.5 bar for 20 min to achieve a stabilized state, and then the performance was measured at 2 bar. The dye concentrations (MB, CR, CB-T, AF, RB) in the feed solution were kept as 0.1 g/L.

Water permeance (F, L·m^−2^·h^−1^·bar^−1^) was calculated by Equation (1)
(1)$$F=V/\left(A*t*\Delta P\right)\;\;$$
where V, t, A, and ΔP are the volume of the permeated solution (L), filtration time (h), effective membrane area (m^2^), and transmembrane pressure (bar), respectively.

The rejections for dyes were calculated by Equation (2)
(2)$$R=\left(1-C_{P}/C_{F}\right)\;*100\%$$
where $C_{P}$ and $C_{F}$ are the dye concentrations in the feed and permeate solutions, respectively.

## 3. Results and Discussion

### 3.1. Characterizations of HC6H

The 1H solid-state NMR and FT-IR spectra of HC6H powder were obtained to confirm its chemical structure. As shown in Figure 3a, the observed peaks in the 1H NMR spectrum w are consistent with the previously reported results [11]: δ = 1.23 (s, 2H), 1.33 (t, 2H), 2.58 (t, 2H), 2.95 (q, 2H), 3.21 (q, 2H), 5.78 (t, 1H), 5.88 (t, 1H), 6.77 (s, 1H), 7.53 (s, 1H), 11.85 (s,1H). As shown in Figure 3b, the FT-IR spectrum presents characteristic peaks at 1240 cm^−1^ (-C-C-), 1616 cm^−1^ (-NH-), 1736 cm^−1^ (-C=N-), and 2847 cm^−1^ and 2921 cm^−1^ (-CH2-), further confirming the successful synthesis of HC6H.

The crystal structure of HC6H powder was characterized by XRD, as shown in Figure 3c. The main diffraction peaks are located at 2θ = 4.09° (100), 8.27° (200), 12.38° (300), and 26.99° (001), indicating a moderate crystallinity of the obtained HC6H powder. According to the Bragg equation [21], the minimum interlayer spacing is 3.3 Å, which could transport water and reject the dyes. The structure of HC6H powder was determined by TEM, and the results are shown in Figure 3d,e. It can be clearly seen that the HC6H exhibits a planar and ordered layered pore structure, which matches the single crystal structure in the XRD pattern. Meanwhile, the pore structure of HC6H is stacked in an ordered layered structure (Figure 3d), which matches the periodic results of the layered thin sheets in the XRD pattern, indicating the I-quartet crystal structure in the arranged layers of HC6H molecules. When the ethanol solution of HC6H was poured onto the surface of PAN substrate, the self-assembly process occurred, and the resulting pore structure provides sufficient channels for water molecules. In the next step of interfacial polymerization, the COF layer was in contact with the water channel, and a composite structure was successfully formed onto the membrane surface.

TGA was used to explore the composition and thermal stability of HC6H (Appendix A). A small amount of mass loss is exhibited in the temperature range of 100–160 °C because of the strong binding of water molecules to the channels. The powder firstly decomposed at 180 °C and the weight loss could reach 74.19% at 180–440 °C. The secondary decomposition starts at 440 °C, and the weight loss at this stage was about 7.97%, which is consistent with the previous report [11]. Meanwhile, it also indicated the HC6H powder can maintain a stable self-assembled structure at 180 °C, and its thermal stability can meet the requirements for dye separation and wastewater treatment.

### 3.2. Morphologies of Composite Matrix Membranes

Appendix A display the SEM and AFM images of the top surface of composite matrix membranes fabricated with different reaction times and HC6H concentrations. The variations of surface roughness with HC6H concentrations and reaction times were shown in Figure 4a,b.

It can be seen that both PAN and HPAN membranes display smooth surfaces; the formation of the COF layer results in a rough and dense surface (Appendix A). When the HC6H concentration was adjusted in a small range (≦2.0 mg/mL), it had little effect on the surface morphology of the membrane (Appendix A), and the surface roughness value was stable in the range smaller than the pristine membrane (Figure 4b). As the HC6H concentration increased above 2.0 mg/mL, granular substances appeared on the membrane surface (Appendix A) and surface roughness increased (Figure 4b). This is because the reaction sites were saturated and a large amount of HC6H accumulated on the membrane surface. The accumulation of HC6H will inevitably lead to a decrease of permeance.

As the reaction time increased from 10 to 25 min, the surface roughness values first increased and then decreased (Figure 4a), which is consistent with the SEM images. When the reaction time was short (10 min), the membrane surface had obvious stripes and high roughness due to the rapid formation of the COF layer (similar to Appendix A). When the time was extended to 15 min (Appendix A), water channels formed by HC6H appeared and the surface became smooth. When the reaction time was extended to 25 min, the strip-like protrusions disappeared and the film surface was smoother. However, when the reaction time reached 30 min, the formed cross-linked structure blocked the membrane pores, so a granular structure (Appendix A) with a surface roughness of 12 nm (Appendix A) on the membrane surface could be clearly seen. As the reaction time was extended to 20 min, the strip-like protrusions on the membrane surface had disappeared, indicating the formation of a good planar configuration (Appendix A).

### 3.3. Surface Hydrophilicity of Composite Matrix Membranes

Surface hydrophilicity plays a critical role in improving the permeability of the membrane. Figure 5a shows water contact angles of the membranes fabricated with different reaction times and HC6H concentrations. The water contact angles of the PAN substrate decreased from 60.0 ± 1.5° to 37.2 ± 1.5° after the hydrolysis in sodium hydroxide (Appendix A). After the formation of TpPa layer, the surface of TpPa/HPAN membrane displayed a water contact angle of 45.9 ± 1.5°. After incorporating 1.0 mg/mL of HC6H, the HC6H−TpPa/HPAN membrane surface displayed a water contact angle of 53.5 ± 1.5°, which might be due to the long hydrophobic chain alkanes of HC6H. When the HC6H concentration was above 1.0 mg/mL, the water contact angle of the membranes decreased continuously, probably due to the synergistic effect of hydrophilic groups and roughness. Both the number of hydrophilic groups and the surface roughness (7.94 nm to 25 nm) on the membrane surface increased with the increase of the concentration.

Figure 5b shows the effect of reaction time on water contact angle. When the reaction time was 10 min, a hydrophilic water channel has not been formed on the membrane surface, and the hydrophobic chain alkanes played a major role on the membrane surface, leading to the increase of water contact angle. When the reaction time reached 30 min, the contact angle dropped to 38.92 ± 1.5°. In general, due to the presence of artificial water channels, the hydrophilicity of the membrane was enhanced and the overall water contact angle was reduced.

### 3.4. Chemical Composition of Composite Matrix Membranes

The chemical structure of the membranes and the molecular interactions between HC6H and TpPa were determined via FT-IR and XPS spectra. Figure 5c compared the FT-IR spectra of HC6H−TpPa/HPAN membrane (red curve), TpPa/HPAN membrane (blue curve), HPAN membrane (green curve), and PAN membrane (black curve). The FT-IR spectrum HPAN exhibited adsorption bands at 1569 cm^−1^ due to the stretching vibration of –COOH compared with the spectrum of PAN; this is because, after the hydrolysis of the PAN substrate in sodium hydroxide solution, the cyano group of PAN was converted to a carboxyl group. The FT-IR spectrum of HC6H−TpPa/HPAN exhibited adsorption bands at 1230 cm^−1^ and 1650 cm^−1^ due to the stretching vibration of -C-C- and -C=N-, respectively, indicating the successful assembly of artificial water channels on the membrane surface. In addition, the adsorption bands at 3354 cm^−1^, assigned to -OH group, indicated the existed interaction between TpPa layer and artificial water channels.

As demonstrated in Figure 5d, XPS measurement was used to explore the chemical bonding and surface elements of TpPa-/HPAN and HC6H−TpPa/HPAN membranes. Both membranes showed three peaks at 285.3 eV (C 1 s), 398.68 eV (N 1 s), and 530.98 eV (O 1 s). Appendix A lists the elemental atomic percentage of carbon (C), oxygen (O), and nitrogen (N) on the membranes surface. Compared with TpPa-/HPAN membrane, the HC6H−TpPa/HPAN membrane showed a decreased O content and an increased N content, with the N/O ratio increased from 2.16 to 2.97. This might be because the HC6H exhibited a higher nitrogen element, with an N/O ratio of 3.5, than the TpPa layer. Therefore, the increase of N/O ratio also proved the successful incorporation of HC6H into the membrane. According to the change of N/O ratio in the membrane, we can calculate that the proportion of HC6H in the composite membrane was 60.4%.

### 3.5. Seperation Performance of Composite Matrix Membranes

The key reaction parameters in the process of HC6H self-assembly were investigated to optimize the separation performance of HC6H−TpPa/HPAN membrane, including reaction time, HC6H concentration, and reaction temperature.

In the process of membrane preparation, the HC6H concentration was a critical factor determining the generation of artificial water channels. The influence of HC6H concentration on water permeance and CR rejection was investigated, and the result is shown in Figure 6a. It can be seen that the water permeance of the pristine membrane was around 135.8 L·m^−2^·h^−1^·bar^−1^. As HC6H concentration increased from 1.0 mg/mL to 2.0 mg/mL, water permeance dramatically increased from 169.8 to 271.7 L·m^−2^·h^−1^·bar^−1^. These results indicated that a higher HC6H concentration is beneficial to the formation of water channels in the membrane. However, further increasing the HC6H concentration to 3.0 mg/mL caused a significant decline in water permeance to 216.5 L·m^−2^·h^−1^·bar^−1^. As the assembly of HC6H in the membrane tended to be saturated, excessive HC6H stayed on the membrane surface or in the channels, leading to a decrease in water permeance. Furthermore, the rejection rate to CR of all the composite matrix membranes was above 99.9%, and the maximum water permeance was almost twice that of the pristine membrane. As a result, 2.0 mg/mL HC6H was chosen to be the optimal concentration for the following investigation.

Similar to the self-assembly of HC6H in other substrates [22], the reaction time with HC6H has a great influence on the membrane performance. It directly determines the number of artificial water channels formed in the membrane. Figure 6b displays the pure water permeance and CR rejection of the membranes fabricated with different reaction times. It can be seen that the water permeance of the TpPa/HPAN membrane was 135.8 L·m^−2^·h^−1^·bar^−1^. As the reaction time increased from 10 to 25 min, water permeance of the HC6H−TpPa/HPAN membranes increased significantly from 225 to 297 L·m^−2^·h^−1^·bar^−1^. When the reaction time was relatively short, most of the HC6H had not yet assembled into the water channel structure, but more and more water channels were formed in the membrane with the increase of self-assembly time. When the reaction time reached 30 min, water permeance reduced to 244 L·m^−2^·h^−1^·bar^−1^. This might because the self-growth of HC6H, with a longer reaction time, easily led to the formation of a cross-linking structure, leading to a pore blockage. The CR rejection of all composite matrix membranes was higher than 98.7%. Based on the above results, 20 min was chosen as the optimal reaction time with HC6H.

Finally, the effect of reaction temperature on the separation performance of the membrane was also studied (Figure 6c). When the reaction temperature increased from 20 °C to 60 °C, the water permeance of the membrane increased slightly from 271.7 to 286.6 L·m^−2^·h^−1^·bar^−1^, and the CR rejection rate was still maintained at a high level (>99.9%). According to the Arrhenius law [23], an appropriate increase in temperature is conducive to the formation of more water channels within a certain period of time. Since the effect of temperature on the membrane properties is not obvious, 20 °C was chosen as the self-assembly temperature.

After investigating the optimal preparation parameters of the composite matrix membranes, we further investigated their separation performance using several typical dyes (Figure 6d). It can be seen that the rejection rates of the membrane for CR (2.85 × 0.89 nm), AF (1.17 × 1.13 nm), CB-T (1.55 × 0.88 nm), MB (1.25 × 0.51 nm), and RhB (1.69 × 0.83 nm) were 99.9%, 94.3%, 97.7%, 95.8%, and 96.5%, respectively, indicating that the composite matrix membrane had excellent separation performance. Owing to the formation of artificial water channels in the membrane, they can effectively repel these five dye molecules (rejection rate > 90%) with high water permeance.

### 3.6. Stability of Membrane

The mechanical stability of composite matrix membranes was investigated by measuring the water permeance under different operating pressures. As shown in Figure 7a, when the pressure increased from 1.0 to 3.5 bar, the water permeance increased almost linearly from 191.08 to 360 L·m^−2^·h^−1^·bar^−1^, indicating that the water channels of composite matrix membranes remained at good rigidity below this pressure.

In addition, the composite matrix membranes were immersed in NaCl solution (2.0 mg/mL) or aqueous alcohol for 7 days, and then their dye rejection and water permeance were measured to reveal chemical stability. As shown in Figure 7b, the water permeance changed little, and the dye rejection remained at a high level, indicating that the NaCl solution and aqueous alcohol could hardly destroy the membrane structure. According to the above results, the composite matrix membranes have potential for use in the application of dye separation from saline and organic wastewater.

In order to highlight the superior performance of the composite matrix membranes, we compared its dye rejection and water permeance with other membranes (Figure 8). Many reported membranes exhibited a moderate water permeance (<50 L·m^−2^·h^−1^·bar^−1^) with a high RhB dye rejection rate (e.g., >95%), or a high water permeance (>200 L·m^−2^·h^−1^·bar^−1^) with a low RhB rejection rate (<90%) [24]. In contrast, the HC6H−TpPa/HPAN composite matrix membranes, fabricated in this work, exhibited a water permeance of 271 L·m^−2^·h^−1^·bar^−1^, and rejection rates to several dyes above 94%. Such an excellent water permeance should be ascribed to the remarkable composite structure composed of COF layers and HC6H.

## 4. Conclusions

In this study, we presented a composite self-assembly strategy to prepare advanced COF membranes by introducing artificial water channels. Using the hydrogen bonding between HC6H molecules and COFs, a highly efficient artificial water channel structure in the membrane was successfully constructed. The hydrophilicity and water permeability of the membrane surface have been greatly improved by the introduction of artificial water channels, it breaks the permeability and selectivity trade-off effect, and it has a high separation performance in dye rejection. Under optimized synthesis conditions, the composite matrix membrane exhibits excellent water permeance (271.76 L·m^−2^·h^−1^·bar^−1^) and high rejection rates (>94%) to different dye molecules. Moreover, the membrane stability was improved due to the multiple interactions between the water channels and COF layers. Additionally, the composite matrix membranes showed stable permeation and rejection performance under different conditions; the good thermal stability of the water channel structure also enables the composite membrane to meet the higher requirements of dye separation. It follows that, using artificial water channels to design composite structures, as in this study, may prove to be an effective strategy to optimize the separation performance of the COF membrane. It shows great potential in the treatment of dyeing wastewater.

## Figures and Tables

**Figure 1 membranes-12-00824-f001:**
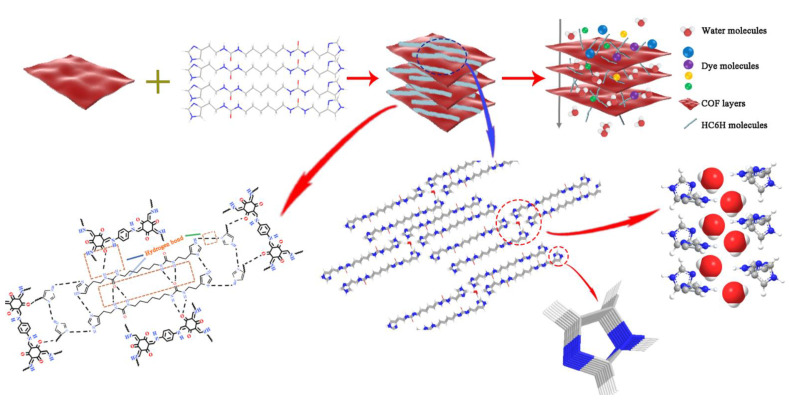
The composite architecture of TpPa-HC6H.

**Figure 2 membranes-12-00824-f002:**
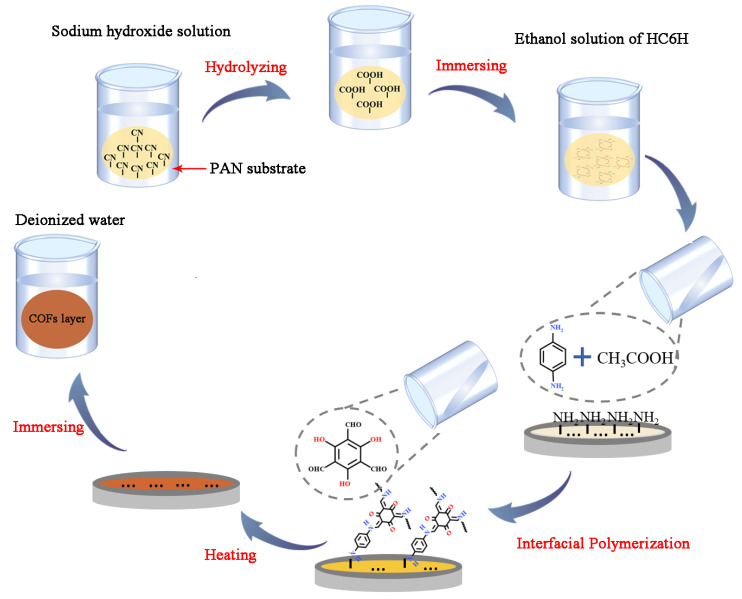
The synthesis process of composite matrix membranes by interfacial polymerization.

**Figure 3 membranes-12-00824-f003:**
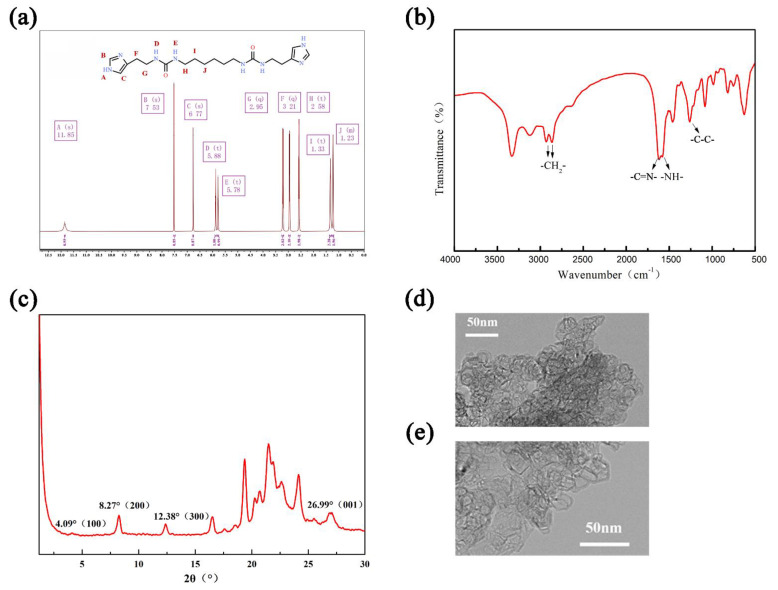
The characterizations of HC6H powders. (**a**) ^1^H NMR spectroscopy of the synthesized HC6H powders. (**b**) FT−IR spectra of HC6H powders. (**c**) XRD pattern of HC6H powders. (**d**) TEM pattern of HC6H powders (60 k). (**e**) TEM pattern of HC6H powders (80 k).

**Figure 4 membranes-12-00824-f004:**
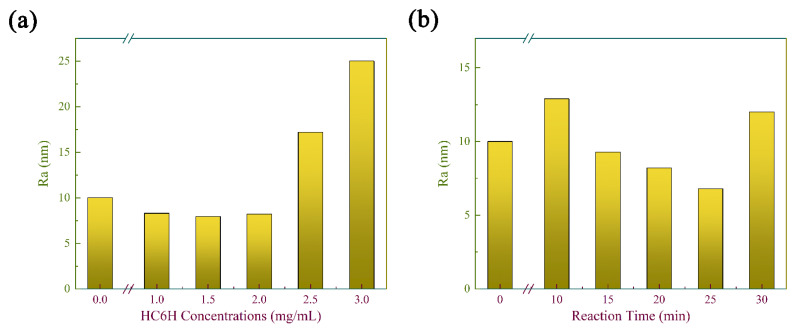
The effect of HC6H concentration (**a**) and reaction time (**b**) on surface roughness of HC6H−TpPa/HPAN membrane.

**Figure 5 membranes-12-00824-f005:**
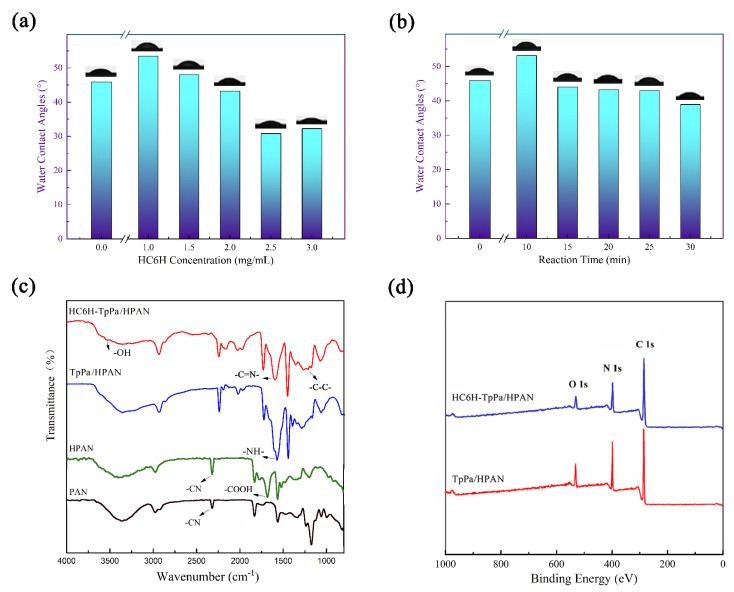
The influence of reaction concentration (**a**) and reaction time (**b**) on water contact angle of HC6H−TpPa/HPAN. (**c**) FT−IR spectra of HC6H−TpPa/HPAN membrane (**red curve**), TpPa/HPAN membrane (**blue curve**), HPAN membrane (**green curve**), and PAN membrane (**black curve**). (**d**) XPS spectra of TpPa/HPAN membrane and HC6H−TpPa/HPAN membrane.

**Figure 6 membranes-12-00824-f006:**
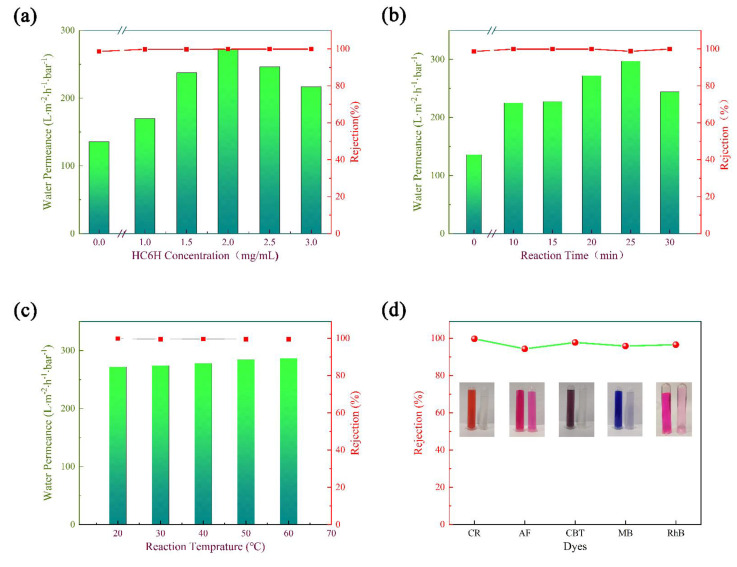
The influence of reaction concentration (**a**), reaction time (**b**) and reaction temperature (**c**) on the separation performance of the HC6H−TpPa/HPAN membranes. (**d**) The rejection performance of HC6H−TpPa/HPAN membranes to different dyes (CR, AF, CB−T, MB, and RhB).

**Figure 7 membranes-12-00824-f007:**
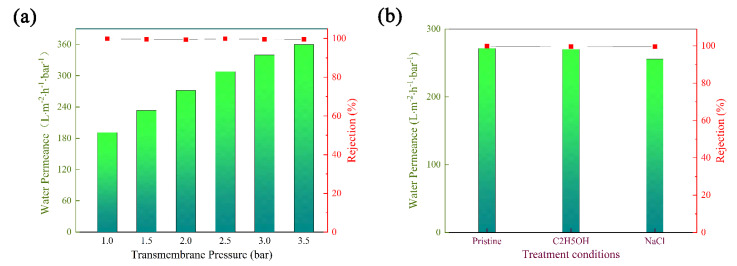
(**a**) Water permeance under different transmembrane pressures on composite matrix membranes. (**b**) Separation performance before and after different solution treatments for the HC6H−TpPa/HPAN membranes.

**Figure 8 membranes-12-00824-f008:**
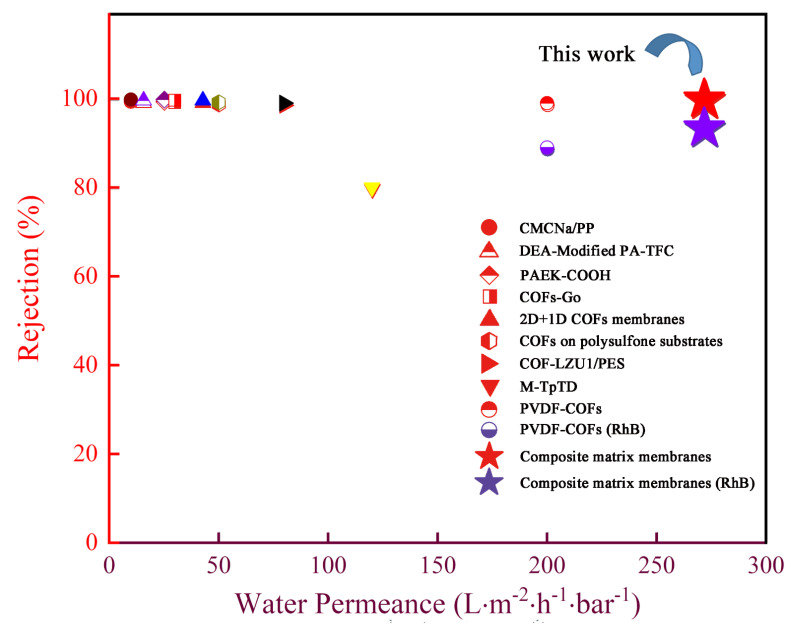
Comparation of dye rejection (red: CR, blue: RhB) and water permeance between HC6H−TpPa /HPAN composite matrix membranes and other membranes reported in the literature. Detailed information for these membranes is given in Appendix A.

## Data Availability

Not applicable.

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
