# Peer review of "High-Flux Ultrafiltration Membranes Combining Artificial Water Channels and Covalent Organic Frameworks"

_membranes, 2022, doi:10.3390/membranes12090824_

Round 1
Reviewer 1 Report
The authors have produced using an interfacial polymerization and self-assembly technique that involved the use of polyacrylonitrile ultrafiltration materials modified with synthetic water channels (HC6H). The combination of COFs layers and imidazole-quartet water channels gives the membrane excellent performance. The composite structure based on AWCs and COFs may provide a cutting-edge framework for the development of separation processes, and the membranes have a high potential for removing dyes from sewage. I believe this paper can be published after making the necessary improvements.
The following requirements must be met by the study's authors and researchers:
1. Please correct almost all of the English in the manuscript and, if necessary, send it to the proofreader before resubmitting.
2. Please increase the resolution of all figures in the manuscript.
3. This statement should be improved to make it clearer (line 302-307) “As the reaction time increased from 10 to 25 minutes, water flux increased significantly from 225 to 297 L·m-2·h-1·bar-1. When the reaction time was relatively short, most of the HC6H have not yet assembled into water channel structures, but more and more water channels were formed in the membrane with the increase of self-assembly time. When the reaction time achieved 30 minutes, the water flux reduced to 244 L·m-2·h-1·bar-1”.” Why only 10 to 25 minutes for a reaction time? not tested for the performance of this membrane for 5 minutes, 45 minutes, or 1 hour.
4. Please raise the manuscript's conclusion.
Author Response
We are very grateful for your constructive suggestions. According to your suggestions, we have revised manuscript. The revised contents have been highlighted in the revised manuscript.
Point 1: Please correct almost all of the English in the manuscript and, if necessary, send it to the proofreader before resubmitting.
Response 1: Thanks for your valuable advice. In order to improve our manuscript, we have consulted the professional language editing services. Under their help, the grammatical and typographical errors have been corrected, and the expression in our manuscript has been well improved.
Point 2: Please increase the resolution of all figures in the manuscript.
Response 2: Thank you very much for your suggestions. the resolution of all figures in the manuscript have been increased. 。
Revisions in Manuscript:
Please see the all figures in the revised manuscript.
Point 3: This statement should be improved to make it clearer (line 302-307) “As the reaction time increased from 10 to 25 minutes, water flux increased significantly from 225 to 297 L·m-2·h-1·bar-1. When the reaction time was relatively short, most of the HC6H have not yet assembled into water channel structures, but more and more water channels were formed in the membrane with the increase of self-assembly time. When the reaction time achieved 30 minutes, the water flux reduced to 244 L·m-2·h-1·bar-1”.” Why only 10 to 25 minutes for a reaction time? not tested for the performance of this membrane for 5 minutes, 45 minutes, or 1 hour.
Response 3: We thank the reviewer for raising this question, We have carried out experiments with HC6H reaction time of 5 minutes, 10 minutes, 15 minutes, 20 minutes, 25 minutes, 30 minutes, 45 minutes, 60 minutes and 90 minutes respectively. When the reaction time is 5 minutes, the addition of HC6H has a very small improvement on the performance of the membrane (Water permeance: 150 L·m-2·h-1·bar-1). When the reaction time exceeds 30 minutes, the water flux of the membrane will stabilize at a relatively low level, about 160 L·m-2·h-1·bar-1. Considering comprehensively, the reaction time of subsequent experiments is 10 to 30 minutes and no longer or shorter reaction time was explored.
Point 4: Please raise the manuscript's conclusion
Response 4: Thank you very much for your suggestions. We have revised the conclusion of the manuscript’s.
Revisions in Manuscript:
In this study, we present a composite self-assembly strategy to prepare advanced COFs membrane by introducing artificial water channels. Using the hydrogen bonding between HC6H molecule and COFs, a highly efficient artificial water channel structure in the membrane was successfully constructed, the hydrophilicity and water permeability of the membrane surface have been greatly improved by the introduction of artificial water channels, it breaks the permeability and selectivity trade-off effect, and has a high separation performance in dye rejection. Under optimized synthesis conditions, the composite matrix membrane exhibits excellent water permeance (271.76 L·m-2·h-1·bar-1) and high rejection rates (>94%) to different dye molecules. Moreover, the membrane stability has been improved due to the multiple interaction between the water channels and COFs layers. Besides, the composite matrix membranes showed stable permeation and rejection performance under different conditions, the good thermal stability of the water channel structure also enables the composite membrane meet the higher requirements of dye separation. It follows that using artificial water channels to design composite structure in this study may be an effective strategy to optimize separation performance of the COFs membrane. It shows great potential in the treatment of dyeing wastewater.

Reviewer 2 Report
The manuscript membranes-1840298 entitled "High-flux membranes based on the perfect combination of Artificial Water Channels and Covalent Organic Frameworks" focused on the fabrication of composite ultrafiltration membranes using PAN/alkyl ureido-imidazole (HC6H)/ COF structures and performed their characterizations. Further, water flux and dye rejection rates of the membranes were tested. The hydrophilicity of the membrane has improved after modification and showed higher water flux and higher rejection rates. Overall, the work is well-written and flows well. Although the work is interesting, it needs further improvements for the publication in Membranes. Thus, I recommend publication after major revision. Some specific suggestions for the authors are:
1- Although HC6H is soluble in ethanol, authors have used solid-state HNMR spectroscopy to obtain the spectrum. Could authors inform the reader about the solubility of these materials at the beginning of the discussion and explain why solid NMR was chosen.
2- TEM images presented in Fig. 3d and e do not provide any magnification scale. What are the magnifications in those images?
3- Hydrolysis of PAN may result in an amide product instead of carboxylic acid. How did the authors confirm the chemical structure after hydrolysis? Characterization of HPAN is imperative as it may directly affect the following modification.
4- Could the authors provide SEM image of PAN and HPAN membrane without further reactions? It would be more interesting if the author could provide digital images of the membranes during each step.
5- Please provide better SEM images (Fig. S4) as the pictures do not show a significant difference at this magnification.
6- What are the actual reactions between substrate and additives? Is HC6H reacting with Tpa? If not, could the authors explain how it remains stable after washing it? Or how can authors make sure it does not leach after filtration?
7- The authors did not mention how much HC6H was integrated into the membrane matrix. Could you quantify the amount of HC6H by comparing TPA modified hydrolyzed PAN sample? Or how much of the membrane was actually modified with COF or HC6H?
8- Considering the surface hydrophilicity? What are the error bars for each sample? Could the authors explain how many measurements have been performed to obtain these results?
9- On page 7, line 232, the authors mentioned; “Compared with the pristine membrane (45.9°)….” What authors actually refer to pristine membrane? Is it PAN or hydrolyzed PAN or TPA modified PAN? Have authors tested the contact angle of PAN and HPAN? After hydrolysis, the contact angle must be reduced too.
Author Response
We are very grateful for your constructive suggestions. According to your suggestions, we have made supplementary experiments and analysis, and revised manuscript. The revised contents have been highlighted in the revised manuscript.

Round 2
Reviewer 1 Report
I recommend it be published in this form. All of the necessary revisions have been completed.
Author Response
Thank you very much. We are very grateful for your recognition of our work.
Reviewer 2 Report
Unfortunately, no significant improvement has been made in the revised manuscript. The previous suggestions were the points that should be integrated in the manuscript not just to respond so that future readers will not have any concern about the study. I recommend authors to improve the discussion based on the previous comments.
Author Response
Sorry about that. We have made supplementary experiments and analysis, and revised manuscript. The revised contents have been highlighted in the revised manuscript.

Round 3
Reviewer 2 Report
The authors made substantial improvements, therefore, I recommend the manuscript for publication.